# Serum Creatine Kinase-MB Isoenzyme Activity among Subjects with Uncomplicated Essential Hypertension: Any Sex Differences

**DOI:** 10.3390/medsci5020008

**Published:** 2017-04-27

**Authors:** Mathias Abiodun Emokpae, Goodluck O.N.A. Nwagbara

**Affiliations:** 1Department of Medical Laboratory Science, University of Benin, Benin City 300283, Nigeria; gonnwagbara@gmail.com; 2Defence Reference Laboratory, Health institution, Abuja-Nigeria, FCT-Abuja 900211, Nigeria

**Keywords:** cardiovascular events, creatine kinase-MB isoenzyme, hypertension, troponin I

## Abstract

Hypertension (high blood pressure) is a major health challenge and more women than men are affected by the condition. Complications as a result of this condition often lead to disabilities and premature death. The objective of this study was to evaluate creatine kinase-MB (CK-MB) activity in uncomplicated hypertension and to know whether sex differences exist in the activity of the enzyme. Serum creatine kinase-MB isoenzyme activity, troponin I, and lipid profile were evaluated in 140 male and 100 female Nigerians with hypertension. The control group was comprised of 100 (50 males and 50 females) normotensive subjects. Measured parameters were assayed using Selectra Pros chemistry analyzer. The means were compared between males and females using Students’*t*-test. The mean CK-MB activity of the female hypertensive subjects was significantly higher (*p* < 0.001) than the males. Similarly, the mean cardiac troponin I (cTnI) of the female hypertensive subjects was significantly higher (*p* < 0.001) than the males. Conversely, the mean CK-MB activity of the female normotensive subjects was significantly lower (*p* < 0.001) than the male counterparts. There was no difference in the levels of cTnI between male and female normotensive subjects. Serum CK-MB activity was higher in female than male hypertensive subjects. In the light of these results, cardiac markers should be routinely done in the evaluation of hypertensive subjects and sex-specific consideration may be recognized in the management of these patients.

## 1. Introduction

Hypertension in Nigeria has emerged as a major health problem with enormous socioeconomic importance. It is the commonest single risk factor for cardiovascular related events and deaths [1], which includes stroke, congestive heart failure, chronic kidney disease, and coronary artery disease [2]. The condition is under-detected and under-treated as a result of ignorance and poverty. Therefore, complications could occur that may lead to chronic disabilities and premature deaths [3]. The prevalence of hypertension among Nigerians was estimated to be 15% which may not be a true representation of the burden of the disease [4]. In most countries of sub-Saharan Africa, the prevalence of hypertension was reported to be higher in males than females [5,6,7,8,9,10], but the reverse is the case in Nigeria where the prevalence is higher among females than males [11]. Studies have shown that hypertension was the strongest as well as an emerging risk factor for heart failure and stroke in sub-Saharan Africa [12]. Unfortunately, little or no attention has been paid to laboratory assessment of hypertensive subjects and most of the required biomarkers of disease severity and progression are not routinely available in most health facilities in Nigeria. Studies of organ damage in subjects with hypertension showed that hypertensive heart disease, nephropathy, retinopathy, stroke, and ischemic heart disease are common even at the first contact in health care facilities [13,14,15,16,17].

Cardiac biomarkers such as creatine kinase-MB (CK-MB) isoenzyme, troponin I, and lipid profile are important in the evaluation of hypertensive subjects. Creatine kinase-MB has a sensitivity and specificity of >90% for the detection of myocardial injury. Creatine kinase (CK) is an enzyme of two subunits M and/or B with three different pairs of subunits combine to give rise to three different isoenzymes CK-MB, CK-BB, and CK-MM. The CK-MB is the heart specific isoenzyme and used to be the reference method for the diagnosis of acute myocardial infarction in most laboratories and increased levels are frequently interpreted as objective evidence of myocardial injury clinically [18]. Due to the greater cardio-specifity of cardiac troponin I (cTnI) and cardiac troponin T (cTnT), compared to CK-MB, troponins are increasingly used as diagnostic markers for cardiac injury or infarction [19,20,21]. We previously reported higher levels of serum uric acid among Nigerians with essential hypertension. A significantly raised serum uric acid was observed among 59% of male and 62% of female hypertensive subjects [22]. Some studies have reported sex differences for CK activity resulting in the establishment of different reference ranges for men and women [23]. Several authors have reported that black people of West African descent have higher tissue and serum CK activity than Caucasians [24]. It was suggested that high CK activity in black population may be responsible for increased risk of hypertension, since the enzyme aids highly energy-demanding processes in the body such as sodium retention, cardiovascular contractility, and modulation of arteries [24,25,26,27]. Consequently, high CK activity especially in resistance arteries may enhance pressure responses and increase blood pressure [24,25]. It was reported that the impact of misclassification of subjects in clinical or epidemiological studies may be challenging and even misleading if sex differences in CK-MB activity are not taken into consideration [25]. Therefore, analysis of sex differences of some biomarkers of cardiovascular risk events may be important in the management of hypertension. This study evaluates CK-MB activity in uncomplicated hypertension and aims to know whether sex differences exist in the activity of the enzyme among Nigerians with uncomplicated hypertension.

## 2. Patients and Methods

This prospective case controlled study of 240 hypertensive subjects made up of 140 males aged 48.3 ± 1.15 years; confidence interval (CI): 34–62 years and 100 females aged 44.7 ± 1.7 years; CI: 28–61 years. The control subjects were 100 normotensive subjects made up of 50 males aged 41.7 ± 1.7 years; CI: 29–54 years and 50 females aged 40.0 ± 1.40 years; CI: 28–61 years. The mean systolic blood pressure (SBP) of the male hypertensive subjects was 148.02 ± 2.28 mmHg; CI: 121–174 mmHg and for female hypertensive subjects was 152.16 ± 2.33 mmHg; CI: 129–175 mmHg. The mean diastolic blood pressure (DBP) for the males was 95.32 ± 1.70 mmHg; CI: 75–115 mmHg and for females was 96.23 ± 1.12 mmHg; CI: 85–107 mmHg. The study participants were Nigerians with uniform ethnic background. Among the study participants, 40 males and 30 females were either on nefidepine and/or lisinopril anti-hypertensive treatment.

### 2.1. Ethical Consideration

The study was approved by Ministry of Defense Health Research Ethics Committee, Abuja-Nigeria (code MODHREC APP/045 dated 3 July 2015) and participants gave informed consent before specimens were obtained.

### 2.2. Inclusion and Exclusion Criteria

Study participants were hypertensive patients above 18 years of agewho were attending clinic at Defense Headquarters Medical Clinic, Mogadishu Cantonment, Asokoro, Abuja. Those subjects below 18 years and/or diagnosed with hypertension, hypothyroidism, and trauma were excluded from this study. In addition, the subjects who were pregnant or lactating and those who recently donated blood or were vaccinated within the last three months were excluded.

### 2.3. Specimen Collection

Five milliliters of fasting venous blood were collected from the ante-cubital vein. The blood specimens were allowed to clot at room temperature for 30 min and sera were obtained after centrifugation at 1000× *g* for 10min. The sera were used for the determination of CK-MB, cTnI, triglyceride, total cholesterol, and high density lipoprotein cholesterol using Selectra Pro S chemistry auto-analyzer, Puteaux, France and reagents supplied by ELITech group, Rotterdam, Netherlands. The low-density lipoprotein (LDL) cholesterol was calculated using Friedewald’s formula [26]. Serum CK-MB activity was measured serially with a minimum of two values on consecutive days for each subject.

### 2.4. Statistical Analysis

The observed data were normally distributed as tested using W/S test for normal distribution and were analyzed using SPSS version 16 (SPSS Inc., Chicago, IL, USA). Student’s *t*-test was used to compare the means of the male and female participants and level of significance was set at *p* ≤ 0.05. Multivariate regression model was used to test the relationship between CK-MB activity, blood pressure, and sex. Subjects with controlled hypertension (SBP < 140 mmHg and DBP < 90 mmHg who were on antihypertensive drugs treatment) were excluded from analysis of the association between CK-MB, blood pressure and sex.

## 3. Results

As shown in Table 1, the CK-MB activity of the female hypertensive subjects was significantly higher (*p* < 0.001) than the males. Similarly, the mean cTnI of the female hypertensive subjects was significantly higher (*p* < 0.001) than the males. Conversely, the mean CK-MB activity of the female normotensive subjects was significantly lower (*p* < 0.001) than the male counterparts. There was no difference in the levels of cTnI between male and female normotensive subjects. The CK-MB activity was significantly higher in hypertensive than normotensive subjects (Table 2). The multivariate regression analysis of the association between CK-MB activity and SBP (β = 0.19; *p* = 0.02) and DBP (β = 0.16; *p* = 0.01) in women and SBP (β = 0.16; *p* = 0.05) and DBP (β = 0.011; *p* = 0.08) in men was significant. After excluding women (*n* = 30) and men (*n* = 40) on antihypertensive medication, the positive association between CK-MB activity and SBP (β = 18; *p* = 0.03) and DBP (β = 0.17; *p* = 0.04) in women remained significant while a non-significant association of SBP (*p* = 0.09) and DBP (*p* = 0.10) was observed for the males.

## 4. Discussion

The main findings in this study are that CK-MB activity and cTnI levels were significantly higher (*p* < 0.001) in subjects with uncomplicated hypertension than normotensive control subjects. The activity of CK-MB was significantly higher (*p* < 0.001) in female hypertensive than male hypertensive subjects. On the contrary, the activity of CK-MB in female normotensive control subjects was lower (*p* < 0.001) than the male controls. In other words, the sex difference observed in the control subjects was reversed in hypertensive individuals. 

We did not measure total CK and CK-MB mass in this study, which may constitute a limitation in our study. Some authors have reported that measurement of CK-MB using mass may be better than measuring CK-MB activity [27]. Indeed, the guidelines for the redefinition of acute myocardial infarction recommended the use of CK-MB mass rather than CK-MB activity [28]. It was stated that the problems associated with measuring activity include deactivation of the enzyme activity by experimental manipulations leading to underestimation, especially in the low range of activity. In hemolyzed blood samples, adenylate kinase released from the red blood cells and catalyzed the same reaction as CK-MB which could result in false positive results. The presence of CK-BB, macro creatine kinase type 1 or macro creatine kinase type 2 in high concentrations could interfere with the result and lead to false elevation of CK-MB when activity is measured [18]. The fact that CK activity was not assayed after rest may be a limitation, although exercise mainly increases CK-MM activity.

Sex differences in the levels of measured biomarkers may potentially impact negatively on prognosis and clinical outcome of hypertensive women. Therefore, sex should be considered in the management of hypertension among Nigerians. It was previously reported that the incidence of stroke and atherosclerosis was higher in males than females and that women appear protected from cardiovascular diseases until their mid-80s when their incidence of stroke surpasses that of the men [29]. This suggests that there may be sex-specific factors that protect the females from developing hypertensive complications despite higher levels of the measured biomarkers.

Other studies have reported worse prognosis and clinical outcome in women with respect to cardiovascular complications [30,31,32,33,34,35]. The significantly higher CK-MB activity reported in women than men may have been one of the factors responsible for clinical presentation and prognosis of hypertensive women with various cardiovascular complications. Further evaluation is suggested in order to know whether hypertensive Nigerian women develop cardiovascular complications earlier than men. It was reported that women with acute coronary syndrome tend to have worse long-term and short-term than men [30,31,32,33,34,35]. The reason for the sex differences in prognosis was not clear. Whether it was due to different baseline characteristics or due to physiologic distinction between males and females is yet to be ascertained [36]. Meta-analysis of about 35 studies involving 18,555 women and 49,981 men with ST-segment elevation myocardial infarction showed that women have nearly twice the risk for in-hospital all-cause mortality and 1.5 times the risk for one year all-cause mortality compared with men [37].

Sex differences in CK-MB mass were previously studied only in apparently healthy subjects [25,38] and others evaluated the enzyme activity in hypertensive patients [39,40], acute coronary syndrome, and myocardial infarction [41,42,43]. Strunz et al. (2011) reported that females had significantly lower level of CK-MB mass than males among apparently healthy subjects [38]. The differences were attributed probably to the consequence of a greater body muscle mass in men which may indicate the importance of different decision-limit values based on sex in clinical setting. The authors therefore suggested different cut-off values for both males and females [38]. In a study of distribution of CK activity in order to determine eligible subjects for statin therapy, it was stated that about 49% of black respondents had serum CK activity above the upper limit of the reference range. The authors suggested upward review of the upper limit of reference range for use in clinical settings [44]. No study has reported on sex differences in CK-MB activity among hypertensive Nigerian subjects. However, sex differences in total creatine kinase activity have been observed among hospitalized patients with myocardial infarction [45], psychiatric patients [46], and acute coronary syndrome [33]. Several other authors also reported higher levels of total CK in males than females of the same race [47,48,49,50,51,52,53,54,55]. However, no significant different in the means of total CK activity between males and females was reported among Africans which became significant when mean log total CK was compared [54]. Sex differences in CK activity have been reported in the general population and slight elevations associated with increased risk for myocardial infarction especially in individuals with dyslipidemia [55,56].

The higher CK-MB activity in hypertension is consistent with previous reports [41,57]. It was reported that the enzyme activity was elevated in hypertensive subjects with myocardial infarction [57]. The CK-MB activities have been identified as specific and sensitive biomarker of both clinical and subclinical myocardial injury [58], since they are lightly bound to the contractive apparatus and their level in plasma depends on the severity of myocardial injury. The entry of the enzyme in the circulation depends on the rate of passive diffusion of the enzyme from infarct myocardium cells [41,59]. The exact mechanism of minor CK-MB elevation in subjects with hypertension is not completely clear, but most authors believed the leakage of CK-MB may be due to myocardial injury even though controversy does exist [60,61,62]. It was stated that whether associated with pathologically demonstrable myocardial injury or not, mild elevation of CK-MB is prognostically important [41]. Several studies in patients with suspected infarction have shown that even mildly elevated CK-MB activities are associated with worse clinical outcomes [63,64,65,66,67]. However, long-term mortality in those patients was observed to be similar to patients without CK-MB elevation in another study [68]. In studies involving patients with coronary intervention, CK-MB activity was elevated in 10–40% of the patients and was associated with an increased risk of adverse outcomes [69,70,71,72,73,74,75,76,77]. The multivariate regression analysis of CK-MB activity was significantly associated with blood pressure in females after excluding those subjects on antihypertensive medications. It was previously reported that a long-term usage of Captopril (angiotensin converting enzyme inhibitor) may cause cardiovascular disease [78]. A study that evaluated the levels of CK-MB activity in subjects on angiotensin converting enzyme inhibitor and calcium channel blocker observed that CK-MB activity levels correlated positively with duration of drug use. No significant increase was observed in those that had used the drugs for 1–5 years, but in those that had used the drug for 6–12 years, a significantly higher CK-MB activity was observed [40].

In a study that linked blood pressure with high CK activity, it was observed that blood pressure levels increased with each log of CK activity tertile for blood pressure after adjusting for age, sex, body mass index, and ethnicity [79,80]. There was no evidence that CK clearance was changed in those patients with higher blood pressure levels or that circulating CK was derived from the luminal surface of vascular endothelial cells. It was however stated that higher blood pressure levels may have caused cardiovascular muscle damage and increased total CK activity [79]. They did not assay CK isoenzymes but reported that in subjects with uncomplicated hypertension, a normal CK isoenzyme spectrum coupled with relatively high serum total CK activity was observed [80,81]. These authors offered a possible explanation for the relationship between blood pressure and high serum total CK activity at rest which could be due to tissue activity and absence of overt tissue damage or dysfunction.

Studies have suggested that minor elevations in CK-MB are linked with myocardial necrosis [81,82]. Histological data confirmed that elevated CK-MB without an abnormal increase in total CK activity could be linked to several small areas of myocardial necrosis that associated chronologically with the appearance of CK-MB [83]. It was concluded that myocardial necrosis could result from embolization of plaque microparticles, debris of intravascular friable materials, clots, or cholesterol crystals. Others are transient vessel closure, side branch compromise, and coronary dissection. Embolism could also cause small areas of necrosis because of sudden mismatch between metabolic requirement of myocardium and coronary blood flow [43]. High activity of CK-MB might represent a marker of a high-risk population (such as hypertension) with other risk factors that can adversely affect long-term prognosis [84,85].

Mels et al. (2016) observed absence of any link between CK activity and cardiovascular indices in black men and women despite higher CK activities in black women and worse cardiovascular profile of black population compared with Caucasians [23].

## 5. Conclusions

In conclusion, serum CK-MB activity was higher in female than male hypertensive subjects. Sex-specific consideration may be considered in the management of these patients.

## Figures and Tables

**Table 1 medsci-05-00008-t001:** Comparison of measured parameters in male and female hypertensive and normotensive subjects (mean ± SEM).

Measured Parameters	Male hypertensive Subjects (*n* = 140)	Female Hypertensive Subjects (*n* = 100)	*p*-Value	Male Normotensive Subjects (*n* = 50)	Female Normotensive Subjects (*n* = 50)	*p*-Value
Age (Years)	48.3 ± 1.15 (34–62)	44.7 ± 1.7 (28–61)	0.05	41.7 ± 1.70 (29–54)	40.0 ± 1.40 (28–61)	0.50
SBP (mmHg)	148.02 ± 2.28 (121–174)	152.16 ± 2.33 (129–175)	0.6	121.17 ± 2.10 (106–136)	118.25 ± 2.61 (102–136)	0.5
DBP(mmHg)	95.32 ± 1.70 (75–115)	96.23 ± 1.12 (85–107)	0.8	80.50 ± 2.10 (65–90)	71.61 ± 2.16 (56–86)	0.001
cTnI (ng/mL)	0.074 ± 0.001 (0.062–0.086)	0.081 ± 0.001 (0.071–0.90)	0.001	0.001 ± 0.00	0.001 ± 0.00	1.0
CK-MB (U/L)	48.6 ± 1.71 (28.6–68.2)	56.2 ± 1.50 (41–71)	0.001	15.5 ± 0.20 (14.0–16.9)	14.0 ± 0.25 (12.2–15.8)	0.001
Total Cholesterol (mmol/L)	5.45 ± 0.13 (3.92–6.98)	5.83 ± 0.12 (4.61–7.02)	0.01	3.94 ± 0.12 (3.09–4.79)	4.10 ± 0.10 (3.42–4.85)	0.1
Triglycerides (mmol/L)	1.63 ± 0.03 (1.28–1.98)	1.30 ± 0.01 (1.20–1.40)	0.02	0.89 ± 0.04 (0.60–1.17)	0.87 ± 0.04 (0.60–1.15)	0.6
HDL-c (mmol/L)	1.16 ± 0.04 (0.70–1.63)	1.30 ± 0.04 (0.92–1.70)	0.001	1.21 ± 0.05 (0.85–1.56)	1.36 ± 0.05 (1.00–1.71)	0.005
LDL-c (mmol/L)	3.38 ± 0.02 (3.14–3.62)	3.78 ± 0.12 (2.58–4.98)	0.002	2.30 ± 0.10 (1.59–3.00)	2.40 ± 0.07 (1.90–2.89)	0.2

CK-MB = Creatine kinase-MB isoenzyme; cTnI = cardiac troponin I; DBP = diastolic blood pressure; HDL-c = high density lipoprotein cholesterol; LDL-c = low density lipoprotein cholesterol; SBP: systolic blood pressure. Values in parenthesis are confidence intervals.

**Table 2 medsci-05-00008-t002:** Serum activity of CK-MB and troponin I levels in hypertensive and normotensive subjects.

Parameters	Hypertensive Subjects (*n* = 240)	Normotensive Subjects (*n* = 100)	*p*-Values
CK-MB (U/L)	51.6 ± 3.0 (45–58)	15.0 ± 0.75(13–16)	0.001
cTnI(ng/mL)	0.077 ± 0.001(0.055–0.099)	0.001 ± 0.00(0–0.001)	0.001

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
