# Peer review of "Serum Creatine Kinase-MB Isoenzyme Activity among Subjects with Uncomplicated Essential Hypertension: Any Sex Differences"

_medsci, 2017, doi:10.3390/medsci5020008_

Round 1

Reviewer 1 Report

The authors describe CKMB in hypertension.

I have a few comments to improve the paper.

The fact that CK was not estimated after rest is a limitation, which should be acknowledged in the limitation section. Although mainly CKMM increases with exercise, CKMB increases have also been described, but MB% may remain within reference limits.

Women had higher blood pressure and higher CKMB. I am curious to see what this would yield in multivariable regression, does CKMB associate with Blood Pressure independent of sex (and BMI)

Can the authors add a subgroup analysis for total CK. The authors would need to report whether total CK was also increased. Was only CKMB increased or also CKMB as a percentage of total CK?

Based on reference 36 (Amin et al), and 50 (Brewster et al), the authors should report the antihypertensive drugs used and whether patients were controlled or not, and how this might affect CKMB. 

The paragraph on sex differences in CK and CK MB is not well summarized and contains some errors. Please improve, using papers including recent NHANES data on CK; and sex/ethnicity-based reference values in AM H J (PMID 17892987).

Please check your references. Important references are missing (such as Apple et al,
PMID 12881449). Reference 50 and 73 are the same and this reference is wrongly quoted.

Table 1 and 2 should be merged.

The manuscript is well written, but there are a few small typographical errors such as: "Study participants were hypertensive patients above 18years of age"; a missing space between 18 and years. Please edit manuscript for these and other errors.

Author Response

Responses to Reviewer 1

1.       The fact that CK was not estimated after rest is a limitation, which should be acknowledged in the limitation. Although mainly CK_MM increases with exercise, CK-MB increases have also been described, but MB% may remain within reference limits.

Rseponse : This is acknowledged as part of limitation in the discussion section.

2.       Women had higher blood pressure and higher CKMB. I am curious to see what this would yield in multivariable regression, does CKMB associate with blood pressure independent of sex and BMI.

Response: Multivariate regression analysis was done, please see statistical analysis, result and discussion sections.

3.       Can the authors add a subgroup analysis for total CK. The authors would need to report whether total CK was also increased. Was only CK-MB increased or also CKMB as a percentage of total CK?

Response: Total CK was not determined in this study. Only CK-MB activity.

4.       Based on reference 36(Amin et al), and 50 (Brewster et al), the authors should report the antihypertensive drugs used and whether patients were controlled or not and how this might affect CK-MB.

Response: Some of the participants were on either nifedepine or lisinopril this has been adequately addressed in the result and discussion sections.

5.       The paragraph on sex differences in CK and CK_MB is not well summarized and contains some errors, please improve using papers including recent HANES data on CK and sex/ethnicity reference values in Am H J

Response: Additional information added and observed errors corrected.

6.       Please check your references. Important references are missing such as Apple et al. References 50 and 73 are the same and is wrongly quoted.

Response: One of them was removed and the suggested one included.

7.       Tables 1 and 2 should be merged.

Response: Tables merged.

Reviewer 2 Report

The study by Emokpae and Nwagbara details information regarding serum creatine kinase-MB activity in hypertensive Nigerian patients, examining potential gender differences. This study is current and improves the scientific evidence for markers of hypertension in emerging populations. That said, however, it would be important for clarity of the patients to be mentioned up front (i.e. that this is a population of Nigerian individuals) and for this background of individuals to be discussed in the context of global hypertension. Additionally, a few comments relating to the study as a whole remain:

What was the study ratio of different ethnic backgrounds, if any, for the study participants? 

antecedents in the abstract need to be introduced in full upon first mention.

In the study, plasma composition (i.e. triglycerides and lipoproteins) was investigated. In the discussion, it would be advantageous to discuss this in terms of the Nigerian diet composition and how this relates to other world populations and hypertension. 

Finally, the closing of the discussion details the study limitations. The way in which this is presented appears to diminish the findings and could be incorporated earlier on in the discussion. 

Author Response

Response to Reviewer 2

1.      What was the study ratio of different ethnic backgrounds, if any, for the study participants

Response: No ethnic differences, all study participants belong to a uniform ethnic group.

2.      Antecedents in the abstract need to be introduced in full upon first mention

Response: Hypertension (high blood pressure) was added.

3.      In the study, plasma composition was investigated. In the discussion, it would be advantageous to discuss this in terms of the Nigerian diet composition and how this relates to other world populations and hypertension.

Response: We choose to be silent on lipid parameters since diet was not adequate evaluated in this study.

4.      The discussion details the study limitations. The way in which this is presented, it appears to diminish the findings and could be incorporated earlier on in the discussion.

Response: The limitations of the study have been moved upward at the beginning of the discussion section.

Round 2

Reviewer 2 Report

none